# Future Language Modeling from Temporal Document History

**Changmao Li, Jeffrey Flanigan**
University of California, Santa Cruz
{changmao.li,jmflanig}@ucsc.edu

## Abstract

Predicting the future is of great interest across many aspects of human activity. Businesses are interested in future trends, traders are interested in future stock prices, and companies are highly interested in future technological breakthroughs. While there are many automated systems for predicting future numerical data, such as weather, stock prices, and demand for products, there is relatively little work in automatically predicting *textual data*. Humans are interested in textual data predictions because it is a natural format for our consumption, and experts routinely make predictions in a textual format (Christensen et al., 2004; Tetlock & Gardner, 2015; Frick, 2015). However, there has been relatively little formalization of this general problem in the machine learning or natural language processing communities. To address this gap, we introduce the task of **future language modeling**: probabilistic modeling of texts in the future based on a temporal history of texts. To our knowledge, our work is the first work to formalize the task of predicting the future in this way. We show that it is indeed possible to build future language models that improve upon strong non-temporal language model baselines, opening the door to working on this important, and widely applicable problem.[1]

## 1 Introduction

Predicting the future is a standard practice across numerous domains of human life and businesses (Christensen et al., 2004; Tetlock & Gardner, 2015; Frick, 2015). Public and private organizations constantly anticipate future trends, shifts in stock values, or forthcoming technological advancements. The pressure to predict the future has fueled developments in the automated prediction of future numeric data, encompassing areas such as weather forecasting, stock market trends, and demand for goods.

However, it is striking to note the scarcity of work developed towards the automation of predicting textual data. Textual data holds unique significance, given that it is a natural and rich format for human consumption. Moreover, experts frequently offer predictions in a textual format, evident in an array of books, magazines, and academic publications. Despite this, predicting future text data is rarely studied within the machine learning or natural language processing communities.

Our work aims to address this gap by introducing a novel task – future language modeling. The future language modeling task is to construct a generative language model for future text given a temporal history of documents. To the best of our knowledge, this is the first attempt to systematize and advance the task of predicting the future in this specific manner. Beyond formalizing this important task, we also create and develop future language models designed for this task. We evaluate these future language models against strong non-temporal baseline language models using both automatic metrics and human evaluations, and demonstrate their effectiveness at generating future textual content.

A word of caution: predicting the future is a bold claim. We do not wish to argue that all future text can be predicted. There are random events, new named entities, serendipitous discoveries, etc, in text that cannot be predicted. But we hypothesize that there are some important aspects of the future that *can* be predicted given enough historical text. Only by working on this future language modeling task

---

[1] Our code is available at https://github.com/jlab-nlp/Future-Language-Modeling

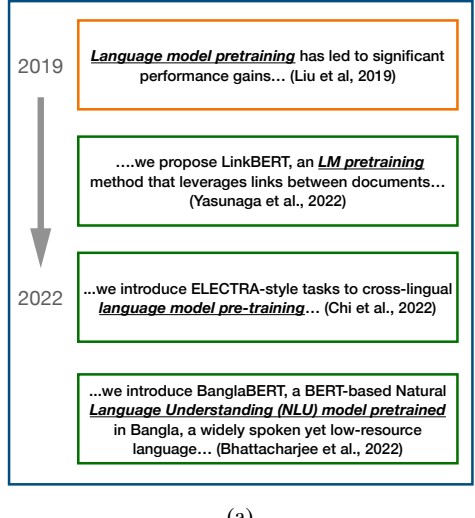 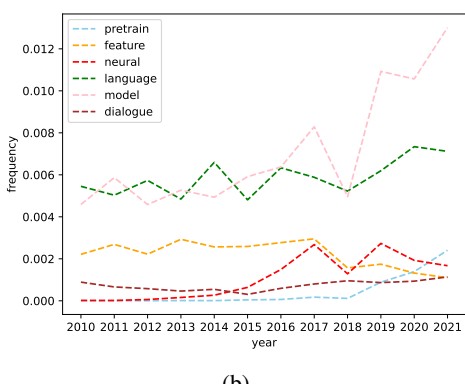

(a)                                                    (b)

Figure 1: (a) represents an example showing how abstracts in recent history are related to the future. In this example, the text of the abstract of the RoBERTa paper (Liu et al., 2019) anticipates the rise of papers about "language model pretraining" (Du et al., 2022; Bhattacharjee et al., 2022; Chi et al., 2022). (b) shows the word frequencies by year in NLP abstracts for some representative words, which reflects topic/approach changes over the years, i.e., "pretrain" started to dramatically go up after 2018 because of BERT, and "neural" became popular after 2013 because of deep learning.

can this hypothesis be verified. We show, by construction, that future language models can be built that perform better, across various automatic and manual evaluations, than non-temporal language models trained on the most up-to-date text, thereby verifying this hypothesis. While humans can sometimes predict the future, experts are often wrong (Frick, 2015), and we do not know the machine upper-bound on this task. We hope to push the boundaries of predicting future trends by proposing the task of and developing methods for future language modeling.

Our contributions are the following:

- We introduce the future language modeling task (§2) of modeling future textual content using a history of documents, as well as evaluation methods for this task (§4.4 & §4.6).
- We develop a series of future language models (§3) for this task that incorporate temporal information into pre-trained language models, which dynamically adjusts the generation probability to generate content that follows the predicted future trend.
- As a concrete example, we evaluate our model to model future abstracts for ACL conference papers, and our proposed approach outperforms the baseline model on both automatic metrics and human evaluation.

The paper is organized as follows. In §5, we present related work. In §2, we provide a task overview to introduce the proposed future text generation task based on texts in previous time spans. In §3, we present the details of our proposed approaches. §4 presents our experiments and results analysis.

## 2 TASK OVERVIEW

We begin by defining some terms. **Without loss of generality, we call the times when we update our language model** *years*, but they could be other time spans such as days or hours. Each year has a collection of texts for that year. For simplicity, we call these texts *documents*.[2]

Our proposed **future language modeling** task is to model future texts using documents from previous years. Let $i$ denote the year index, and document $d_{ij} = \langle x_{ij1}, ..., x_{ijk} \rangle$ be $j$th document from the

---

[2]In our experiments in §4, the texts ("documents") are abstracts.

$i$th year, where $x_{ijk}$ is the $k$th token from the $j$th document in the $i$th year. Let $D_i = \{d_{i1}, ..., dij\}$ represent all documents from year $i$. The task is to generate $D_i$ based on $D_1$ to $D_{i-1}$, which means during generation, the probability of each generated token $x_{ijk}$ is computed not only from a standard language modeling perspective but also considering the content evolution from $D_1$ to $D_i$. The conditional probability for each token $x_{ijk}$ is conditioned not only on the previously generated words in the sentence (as usual), but also on all the previous years' documents:

$$P(x_{ijk}|x_{ij1}...x_{ij(k-1)}, D_1 \ldots D_{i-1}) \tag{1}$$

We call the model for the above task a **future language model**, formally defined as a statistical language model designed to assign high probability to future texts based on the temporal history of texts.

## 3 APPROACH

### 3.1 OVERVIEW OF MODELS

We develop three methods for future language models: a word frequency model (§3.2), a contextual temporal model (§3.3) and a doubly contextualized temporal model (§3.4). In this section, we give some background notation common to all these models.

All our methods modify the language model probabilities to account for the temporal evolution. A language model usually calculates the probabilities with a softmax equation:

$$P(x_k|x_1...x_{k-1}) = \frac{E_{x_k}^T H_k}{\sum_{w'} E_{w'}^T H_k} \tag{2}$$

In this equation, $E_w \in \mathbb{R}^d$ is the learned output embedding vector for the $w$th word in the vocabulary, and $H_k \in \mathbb{R}^d$ is the contextualized embedding at position $k$. We use a transformer language model, and $H_k$ is the vector of the last layer of the transformer decoder in position $k$. This is our baseline to compare with.

Our first two methods (§3.2 & §3.3) compute a temporal bias $B_{iw} \in \mathbb{R}$ for the $w$th word in the $i$th year that is calculated from the previous years. The bias term up-weights or down-weights vocabulary items to account for changes across years. The bias is added into the softmax equation to modify the probabilities:

$$P(x_k|x_1...x_{k-1}, D_1 \ldots D_{i-1}) = \frac{E_{x_k}^T H_k + B_{ix_k}}{\sum_{w'} \left( E_{w'}^T H_k + B_{iw'} \right)} \tag{3}$$

We describe how $B_{iw}$ is calculated in the following sections.

Our third method (§3.4) is more expressive, and calculates a contextualized bias term that depends on the previous words $x_1...x_{k-1}$ that have been generated. This allows the bias term to be contextualized in the output that is being generated. In our notation, the bias term $B_{ikw} \in \mathbb{R}$ is the bias for the $w$th word in the $k$th position in the generated sentence for the $i$th year. The softmax probability equation becomes:

$$P(x_k|x_1...x_{k-1}, D_1 \ldots D_{i-1}) = \frac{E_{x_k}^T H_k + B_{ikx_k}}{\sum_{w'} \left( E_{w'}^T H_k + B_{ikw'} \right)} \tag{4}$$

For training, all our future language models are trained with standard cross-entropy loss:

$$L = -\sum_{k=1}^{|\mathbf{x}|} \log p(x_k|x_1 \ldots x_{k-1}; \theta) \tag{5}$$

where $\theta$ represents the model parameters.

### 3.2 THE WORD FREQUENCY MODEL

Our simplest method models the change over time of the frequency of the words without using any context from historical documents. It only uses the raw counts of the word over time to compute a

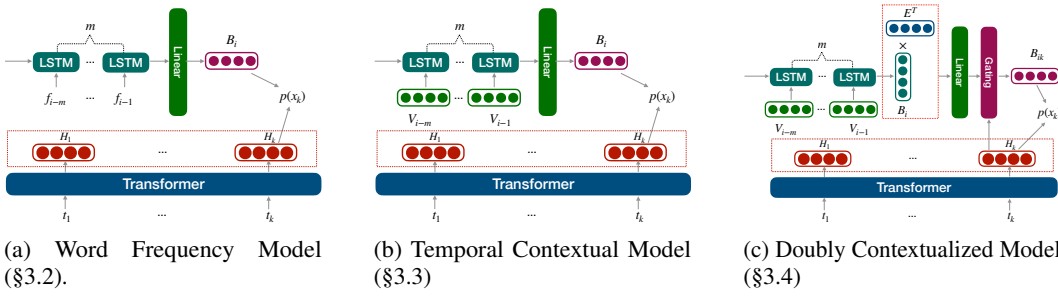

(a) Word Frequency Model (§3.2).

(b) Temporal Contextual Model (§3.3)

(c) Doubly Contextualized Model (§3.4)

Figure 2: Our proposed models.

bias. This bias is added to the final softmax to bias the model towards historical trends. Figure 1b shows the frequency by year for some example words, which reflects topic/approach changes over the years, i.e., the word "pretrain" started to dramatically go up after 2018 because of BERT, and the word "neural" became popular after 2013 because of the deep learning.

Our intuition is to use a temporal neural network to try to predict biases for words based on historical frequency data of words. This model uses an auto-regressive deep learning model to predict the change over time. We use an auto-regressive RNN-style model, specifically an LSTM, rather than a Transformer model for this because it is more naturally suited to our temporal task, as LSTMs do not use position embeddings.[3] For a balance of simplicity, scalability, and expressivity, we use an LSTM.

We predict the temporal word bias for each year using an LSTM and use it as a feature to bias the generation probability. Figure 2a shows the model overview. Let $f_{iw} \in \mathbb{R}$ be the frequencies of the $w$th word for the $i$th year, and let $m$ be the window size which determines how many previous years to consider to predict next year's bias. For each year, we compute a temporal bias $B_{iw} \in \mathbb{R}$ from $m$ previous year's word embedding by using an LSTM where weights are shared across word types. We use the last hidden vector of the LSTM followed by a dot product with a learned vector $A \in \mathbb{R}^d$ to compute the bias:[4]

$$B_{iw} = A^T \, \text{LSTM}(\log(f_{i-m,w}), ..., \log(f_{i-1,w})) \tag{6}$$

This temporal bias is added to the output of the Transformer as a bias in the softmax, as described in §3.1 Equation 3.

### 3.3 THE TEMPORAL CONTEXTUAL MODEL

While the previous method models the change in the frequency of words over time, it does not have contextualized information to help it make its predictions. So while it may see words such as "pretraining" increase over time, it is ignoring contextual information in prior abstracts like "pretraining has led to significant performance gains" that could help it make predictions (see Fig. 1a).

To account for contextualized information contained in prior abstracts, we develop a temporally contextualized model. For each word, we create a pooled representation for each year. We use an average of the contextualized embeddings, averaged over all instances of that word over the year. For each word, we then feed the contextualized embedding into an LSTM to predict the temporal word bias. Figure 2b shows the model overview.

In more detail, using our notation from §2, let $d_{ij} = \langle x_{ij1}, x_{ij2}, ..., x_{ijk} \rangle$ be the $j$th text in $i$th year where $x_{ijk}$ is $k$th token in $d_{ij}$. For the token $x_{ijk}$, let $E_{ijk}$ be the corresponding contextualized vectors from a pre-trained language model. Our representation for the $w$th word in the vocabulary for the $i$th year is the average of the contextualized embeddings, which can be expressed as:[5]

$$V_{iw} = \frac{\sum_{j=1}^{|D_i|} \sum_{k'=1}^{|d_{ij}|} E_{ijk'} I[x_{ijk'} = w]}{\sum_{j=1}^{|D_i|} \sum_{k'=1}^{|d_{ij}|} I[x_{ijk'} = w]} \tag{7}$$

---

[3]We are aware of work demonstrating autoregressive Transformers can be trained without position embeddings, but we leave this style of model for predicting biases to future work.

[4]To make this more efficient, we batch the LSTM across words in our implementation.

[5]We use the indicator function $I[\cdot]$ which is 1 if the condition is true and 0 if it is false.

To use the temporal contextualized word embeddings, we use the fact that more recent years have more influence on future texts and propose a window-sized modeling approach. The window size determines how many previous years for word embedding we consider to predict the next year's temporal bias. Let $m$ be the window size for each year, then we compute a temporal bias $B_{iw} \in \mathbb{R}^d$ from $m$ previous year's word embedding as follows:

$$B_{iw} = A^T LSTM(V_{(i-m,w)}, ..., V_{(i-1,w)}) \tag{8}$$

where we take the last hidden vector of the LSTM and $A \in \mathbb{R}^d$ is a learnable parameter. The temporal bias is added to the output of the Transformer as a bias in the softmax, as described in §3.1 Equation 3.

We also experimented with combining the word frequency model and the temporal contextual model, but we did not observe any improvement by additively combining them.

### 3.4 THE DOUBLY CONTEXTUALIZED MODEL

The temporal contextual model does a good job of predicting the rise and fall in the frequencies of terms. However, we observe that it does a poor job of deciding *when* to use the terms while generating. The `Contextual` output in Table 2 shows an example of this. The model repeatedly introduces new terms that are fashionable, but in an incoherent manner (saying that the paper will focus on IE, but then saying that special attention will be on multi-document summarization).

We hypothesize that the contextual model can predict good terms to use, but cannot decide *when* to rely on the temporal contextual model versus relying on the prior state in the language model (for example, reusing a previous term in the document versus introducing a new fashionable term). The model appears to need a "gating" mechanism to decide when to use the new suggested terms. To address this, we introduce a mechanism that contextualizes the temporal contextual model when generating a document – a *doubly contextualized model* that is contextualized both temporally and in the document generation. Figure 2c shows the model overview.

We start with matching $B_{iw}$ with the pre-trained model embedding space and reduce the dimension of vocabulary size. To implement this, we enable temporal bias $B_{iw}$ to tie with the word embedding layer weights for each word $E_w \in \mathbb{R}^d$ and conduct a linear projection. We compute the tied and projected temporal bias $\tilde{B}_{iw} \in \mathbb{R}^d$ as follows:

$$\tilde{B}_{iw} = (E_w^T B_{iw})A \tag{9}$$

where $A \in \mathbb{R}^d$ is a learnable parameter.

Then we compute the sigmoid attention between transformer decoder output $H_k$ and $\tilde{B}_{iw}$ to obtain the $B_{ikw} \in \mathbb{R}^d$ as follows:

$$B_{ikw} = \alpha \sigma(H_k^T C \tilde{B}_{iw})(E_w^T D \tilde{B}_{iw}) \tag{10}$$

where $C, D \in \mathbb{R}^{d \times d}$ are learnable parameters, and $\alpha$ is a tuned hyperparameter.

This temporal bias is added to the output of the Transformer as a bias in the softmax, as described in §3.1 Equation 4. Using this model, we obtain improved example output shown in Table 2.

## 4 EXPERIMENTS: FUTURE ABSTRACT PREDICTION

### 4.1 DATASET PROCESSING

As a concrete example to be experimented with, we conduct experiments to model future abstracts for NLP papers based on previous papers' abstracts. We first collect paper abstracts for each year from ACL anthology website[6] and filter the noisy abstracts such as papers that are not in English. Then we use the years as the year (for other domains such as news, you can use the day or hour as the year) and split the paper abstracts by years and use abstracts from 2003-2019 as training data, the year 2020 as the development data, and the year 2021 as the test data. Table 1 shows the statistics of the dataset. Figure 3 shows the number of abstracts by year for the dataset.

---

[6]https://aclanthology.org/anthology+abstracts.bib.gz

| Dataset Statistics | Train | Dev | Test |
|---|---|---|---|
| # of abstracts | 37816 | 5919 | 5529 |
| avg. # of sentences per abstract | 9.0 | 6.4 | 6.5 |
| avg. # of tokens per abstract | 225.8 | 168.5 | 164.3 |

Table 1: Data split statistics

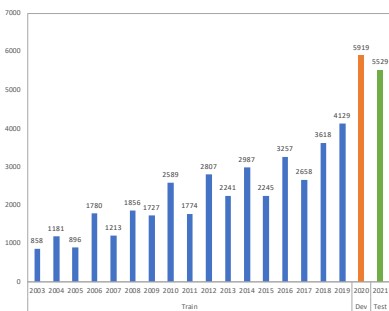

Figure 3: # of abstracts by year

## 4.2  MODELS

We use GPT-2 (Radford et al., 2019) as the pre-trained language model in our experiments, although our approach is not restricted to any particular pre-trained language model. We train and evaluate the following models:[7]

- `Baseline` A baseline which fine-tunes GPT-2 on abstracts from all previous years
- `Baseline-n` A baseline which fine-tunes GPT-2 on abstracts from $n$ most recent previous years, since recent years may be more relevant for predicting future years. We evaluated $n$ from 1 to 10, and report the best 2 models ($n = 2$ and $n = 3$).
- `Frequency-NoLSTM` A word frequency model without using an LSTM, instead directly using the previous year's frequency as a bias feature in the model.
- `Frequency` Word Frequency Model (§3.2)
- `Context` Temporal Contextual Model (§3.3)
- `Context`$^2$ Doubly Contextual Model (§3.4)

## 4.3  HYPERPARAMETER SETTINGS

We use the Adam optimizer (Kingma & Ba, 2015). The batch size is set to 2 with gradient accumulation size 2. Between layers, we apply dropout with a probability of 0.1. We fine-tune 10 epochs for each model and do early stopping. The $\alpha$ is set to $1e - 3$ or initialized with 1 when automatically learned. Bounds for all hyperparameters are the same as GPT-2. We have several hyperparameter search trials on $\alpha$ which are 1, 1e-1, 1e-2, 1e-3, 1e-4, 1e-5. For each model, we have three training and evaluation runs. The method of choosing hyperparameters is based on perplexity scores on the dev set. Fine-tuned RoBERTa Model (Liu et al., 2019) for each year is used to generate temporal word embedding representation. We use beam search decoding with top-k sampling. The beam size is 5, k is 50, and p is 0.92. Since it is topic agnostic, the start token is end of sentence token for GPT-2. All models were trained or evaluated on either one A40 or A6000 GPU. Our implementation is based on Huggingface Transformers (Wolf et al., 2020).

## 4.4  AUTOMATIC EVALUATION METRICS

We use three automatic evaluation metrics, which are perplexity (PPL), content perplexity (CPL), and content meteor (CM). Since most of the evolution of ideas in NLP papers is through changes in

---

[7]Appendix §A gives the model statistics.

| Approach | Examples |
|---|---|
| Baseline | This paper presents a machine learning approach to the task of word sense disambiguation (WSD) based on lexical, syntactic and semantic features. Our approach relies on a rule-based classifier trained on manually annotated corpora. We show that our approach outperforms state-of-the-art WSD systems for English, German and Spanish. |
| Frequency | This paper presents a new approach to the automatic generation of referring expressions (REs) in natural language. The approach is based on the use of distributional semantics, which can be used to model the meaning of an utterance. We describe how we have constructed a data set for RE generation that has been manually annotated with respect to three dimensions: referential quality, fluency, and informativeness. In addition, we show how these dimensions correlate with human judgments about the quality of generated REs. |
| Contextual | This paper presents an overview of the work carried out within the EU FP7 European project Information Extraction (IE). In particular we focus on the main tasks involved in IE, i.e., text classification, named entity recognition and coreference resolution. Special attention will be paid to the issues related to multi-document summarization as well as cross-lingual information extraction which is very important for many natural language processing tasks such as machine translation and question answering. Finally some directions for future research are sketched which include the development of more advanced IE systems based on deep learning architectures and the creation of new IE evaluation corpora. |
| Contextual$^2$ | We present a novel approach to the problem of automatically detecting and classifying semantic relations between named entities (NEs) in natural language text. In particular, we propose a new method for identifying NEs that are not explicitly mentioned in the text, i.e., those that do not appear in training data. The proposed method is based on an unsupervised clustering algorithm that uses word embeddings as features. Experimental results show that our method outperforms state-of-the-art methods by a large margin. |

Table 2: Example output for each approach.

| | Dev | | | Test | | |
|---|---|---|---|---|---|---|
| | PPL↓ | CM↑ | CPL↓ | PPL↓ | CM↑ | CPL↓ |
| Baseline-all | 19.97 | 21.85 | 82.59 | 22.76 | 16.22 | 102.03 |
| Baseline-2 | 21.66 | 24.41 | 91.36 | 21.53 | 19.69 | 94.37 |
| Baseline-3 | 21.08 | 22.75 | 88.75 | 21.06 | 19.01 | 92.06 |
| Frequency-NoLSTM | 19.97 | 22.87 | 83.18 | 21.23 | 19.09 | 88.38 |
| Frequency | 19.97 | 23.87 | 82.43 | 20.20 | 19.98 | 87.68 |
| Context | 23.47 | 24.86 | 96.64 | 23.21 | 18.20 | 102.11 |
| Context$^2$ | **19.66** | **24.94** | **77.54** | **19.81** | **20.12** | **82.43** |

Table 3: Experiments results for automatic evaluation on abstracts. ↓ indicates lower is better and ↑ indicates higher is better. p-value $< 0.001$ for all scores over baseline based on statistical sign test (Dixon & Mood, 1946). Baseline-$n$ means only $n$ previous years' abstracts are used to fine-tune a non-temporal LM. We evaluated $n$ from 1 to 10, and reported the best 2 ($n = 2$ and $n = 3$). Baseline-all means using the whole training set to fine-tune a non-temporal LM. PPL: perplexity score; CM: content meteor score; CPL: content perplexity score (See §4.4 for the detail of these metrics.)

| | Topic | Topic New | Problem | Problem New | Method | Method New | Avg |
|---|---|---|---|---|---|---|---|
| Baseline-all | **100%** | 8% | 50% | **17%** | 42% | 25% | 46% |
| Baseline-2 | 97% | 0% | 60% | 0% | 53% | 3% | 36% |
| Baseline-3 | **100%** | 0% | 95% | **17%** | 35% | 12% | 44% |
| Frequency | **100%** | **17%** | 83% | 0% | **100%** | 25% | 54% |
| Context$^2$ | **100%** | **33%** | **100%** | **17%** | **100%** | **50%** | **63%** |

Table 4: Experiments results for the human evaluation. See §4.6 for details of the criteria.

content words, we manually collect the non-content words as a stopwords list. During content words based evaluation, we filter out the stopwords, and the leftover tokens are naturally formulated into content words. The perplexity score evaluates fluency while the content words based metrics evaluate the adequacy of future research ideas since ideas are mainly represented by content words instead of non-content words.

**Perplexity (PPL)** We evaluate perplexity, which is calculated using the standard formula

$$PPL = 2^{-\frac{1}{M} \sum_i^N \log_2(p(x_i))}$$

where $p(x_i)$ is the token probability computed from the model and $M = \sum_i^N |x_i|$.

**Content Perplexity (CPL)** Perplexity is computed over all words equally, including non-content words. To better evaluate the benefit of the improved content word selection, we calculate perplexity on non-stop words. We call this *content perplexity*. This is computed by ignoring the stopword log probabilities, and only adding the non-stopword log probabilities together and dividing by the number

of non-stopwords instead of the total number of words[8]. For test data $D = x_1, ..., x_N$, the stopwords list is $V_s$, then the content perplexity $CPL$ is computed by

$$CPL = 2^{-\frac{1}{M_s} \sum_i^N \log_2(p(x_i)I[x_i \notin V_s])}$$

where $p(x_i)$ is the token probability computed from the model and $M_s = \sum_i^N |x_i I[x_i \notin V_s]|$

**Content Meteor (CM)** This metric measures the match between model generated abstracts and real abstracts in the dev and test sets. We use 100 random seeds to generate $N_g = 100$ abstracts to compare with all abstracts in the dev or test set. After removing all the stopwords, we evaluate the Meteor score for the generated abstract only by the content words. Let $G = \{a_1, ..., a_i, ..., a_{N_g}\}$ be all generated abstracts with the stopwords removed, let $D = \{d_1, ..., d_j, ..., d_{N_h}\}$ be all abstracts in the dev or test set ($N_h$ is the number of abstracts in dev or test set), we compute Content Meteor as:

$$CM = \frac{\sum_{i=1}^{N_g} \max_{j=1}^{N_h} Meteor(a_i, d_j)}{N_g}$$

### 4.5 AUTOMATIC EVALUATION RESULTS

Table 3 shows the automatic evaluation experiment results across all experimented models. Our proposed methods perform better than the GPT-2 baselines without temporal information on all automatic evaluation metrics.

The doubly contextualized model has about 3 content meteor points improvement over the year agnostic baseline-all, and 5 points content perplexity improvement, which indicates the content better matches real future abstracts. This demonstrates that the doubly contextualized enables the model to generate content words that will be used in the future. The word frequency model also shows 2 points improvement in the content meteor and slightly better in the content perplexity. This indicates that by only adding the word frequency as bias, the model can increase the content matching slightly. We tried the accumulated baselines on the abstracts of the $n$ most recent years and the performance of only training on the most recent abstracts cannot surpass proposed model.

From a fluency perspective, the word frequency model has the same perplexity as baseline-all. In contrast, the doubly contextualized model shows a larger improvement which indicates that it can enable the model to generate more fluent abstracts than the baselines for future abstracts. Without using the LSTM to model the temporal information, the model only considers a single previous year bias which hurts the performance. Without gating, although the model has a high content matching score, it has a lower fluency score because the model cannot recognize which tokens should be biased. This demonstrates the importance of the gating mechanism in the doubly contextualized model.

### 4.6 HUMAN EVALUATION

For a human evaluation, we randomly evaluate 100 generated abstracts for each approach. Since our temporal language generation task is to generate abstracts, we evaluate the abstracts with six different criteria, with criteria tailored to the abstract generation task. Table 4 shows the human evaluation results for all the experimental methods. We have six criteria for evaluation, which are divided into three abstract content types each with fluency and novelty aspects. Each criterion score is binary 0 or 1 for each abstract. We add all obtained scores together and divide them by the total gold scores to obtain the percentage of the human evaluation score. The human evaluators are NLP researchers. We conducted a blind evaluation, so the human evaluators did not know the approach for abstracts.

- **Topic: Is the topic clear and correct?** We check if an abstract has a fluent topic or background description without factual errors.
- **Topic New: Is the topic new?** We check if an abstract has a topic we have never seen before or matches the recent research topics.
- **Problem: Is the problem clear and correct?** We check if an abstract has a fluent problem description without factual errors.
- **Problem New: Is the problem new?** We check if an abstract introduces a problem that we have never seen or matches the recent research problems.

---

[8]Appendix C shows how the stopwords are curated.

- **Method: Is the method clear and correct?** We check if the abstract has a fluent method description without factual errors.
- **Method New: Is method new?** We check if an abstract proposes a method we have never seen or matches the recent approaches.

Note that "new" here does not mean completely new. Instead, it only means more related to "future abstracts" such as abstracts in our dev or test set. Apparently, the model cannot generate completely new topics, a new method, or a new problem that they have never seen during the training.

All of our proposed methods outperform the baseline when evaluated using the average score. The generated topics for all approaches are clear and correct, indicating that the GPT-2 baseline can adequately generate clear topics. However, topics are not necessarily new in the baseline approaches, whereas in our proposed approach 1/3 of the topics are new. Additionally, the problem and the method are not always clear and correct in the baseline, whereas our proposed approaches can have all generated new problems, and the methods are clear and correct. In our proposed approach 1/2 of the approaches are new, which shows that our proposed approaches have the ability to predict new trends for future research.

## 4.7 CASE STUDY

Table 2 shows generated abstracts from all the approaches and Table 6 in Appendix shows more generated abstracts from all the approaches compared to the reference abstracts from ACL conferences. The baseline approach generates more general abstract content that does not contain many details or generate very traditional methods for NLP research. The example from `Context` shows that without gating, the generated output after 2-3 sentences is not related to the starting sentence because it may ignore the previous context, although new content is generated, which shows that the gating mechanism can help the model determine whether the next generated token should be depended on the historical documents or the previous context. In contrast, the `Context`$^2$ method generates more detailed content and content that is more related to recent research, such as word embeddings or neural networks, and later generated sentences are more coherent to the previous context, which balanced between considering the historical documents to generate new content or following the previous context to generate more coherent text.

## 5 RELATED WORK

To the best of our knowledge, there is no prior work constructing language models for future text based on temporal historical documents. However, there is much work on language models with temporal information (Röttger & Pierrehumbert, 2021; Lazaridou et al., 2021; Hofmann et al., 2021; Agarwal & Nenkova, 2022; Loureiro et al., 2022). Huang & Paul (2019) worked on document classification using word-level temporal embeddings, and Röttger & Pierrehumbert (2021) adapts the pre-trained BERT models to domain and time. Lazaridou et al. (2021) evaluated the performance of language models on future text, in a setup similar to ours but did not construct any temporally aware models for future language modeling. Dhingra et al. (2022) conducted experiments with temporal language models for question answering. Hofmann et al. (2021) modeled temporal and social information together by modifying BERT with a latent Gaussian process. Rosin et al. (2022) concatenated time tokens to text sequences and introduced time masking using masked language modeling to make a time-aware BERT. However, none of the previous works are about building language models for future text based on temporal historical documents. In this paper, we fill this gap and propose future language models that can generate texts that are more related to future content, which can be applied to many future forecasting areas.

## 6 CONCLUSION

In this paper, we introduce the task of future language modeling and propose a series of future language models. We evaluate our models on abstracts in NLP. The proposed approaches outperform the baseline non-temporal language models across all automatic evaluation metrics and human evaluation on generating content related to the future text based on temporal historical documents.

## ACKNOWLEDGEMENTS

We thank Nilay Patel, Geetanjali Rakshit, Rongwen Zhao, Brendan King, and Zekun Zhao, and the anonymous reviewers for helpful feedback on earlier drafts. This research was supported by computing resources provided by the Pacific Research Platform's Nautilus cluster, supported in part by National Science Foundation (NSF) awards CNS-1730158, ACI-1540112, ACI-1541349, OAC-1826967, OAC-2112167, CNS-2100237, CNS-2120019, the University of California Office of the President, and the University of California San Diego's California Institute for Telecommunications and Information Technology/Qualcomm Institute, and CENIC for the 100Gbps networks.

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
