|  | TT(h) | GT(s) | MS | ML | HS |
|---|---|---|---|---|---|
| Baseline | 6-8 | 5-10 | 1.2B | 1024 | 768 |
| Frequency | 6-8 | 5-10 | 1.2B | 1024 | 768 |
| Context/Context$^2$ | 7-14 | 8-13 | 1.3B | 1024 | 768 |

Table 5: Model statistics. TT: approximate training time range in hours. GT: approximate generating time range for an abstract in seconds. MS: approximate model size. ML: max length. HS: hidden size

---

**Algorithm 1:** Algorithm to build the temporal word embedding representation

**Result:** The temporal word embedding representation $V$

1 Initialize an empty list $V$ to store vocabulary representation for all years;
2 Initialize year index $i$ to the start year number;
3 **while** $i \leq$ *last year number* **do**
4     Load the finetuned model $M_i$ on year $i$;
5     Load tokenizer $T_i$ for the model;
6     Load abstracts in year $i$ and store them in list $D$;
7     Initialize a list $D_t$ to store tokenized abstracts;
8     **for** $d$ *in* $D$ **do**
9         Append $T_i(d)$ in $D_t$;
10     **end**
11     Initialize a dictionary $V_d$ to store token and token representation pairs;
12     Initialize a counter dictionary $C_d$ to count the number of each token;
13     **for** $d\_ij$ *in* $D_t$ **do**
14         $E_{ij} \leftarrow M_i(d\_ij)$;
15         **for** $k, d_k$ *in* $enumerate(d\_ij)$ **do**
16             **if** $d_k$ *in* $V_d$ **then**
17                 $V_d[d_k] \leftarrow V_d[d_k] + E_{ij}[k]$;
18             **else**
19                 $V_d[d_k] \leftarrow E_{ij}[k]$;
20             **end**
21             $C_d[d_k] \leftarrow C_d[d_k] + 1$;
22         **end**
23     **end**
24     Initialize an empty dictionary $V_i$ to store vocabulary representation for $i$th year;
25     **for** $d_k$ *in* $V_d$ **do**
26         $V_i[d_k] = V_d[d_k]/C_d[d_k]$;
27     **end**
28     Append $V_i$ in $V$;
29     $i \leftarrow i + 1$;
30 **end**
31 **return** $V$;

---

## A   Model statistics

Table 5 shows the model statistics.

## B   The algorithm to build Temporal Word Embedding Representation

Algorithm 1 contains our algorithm to build the Temporal Word Embedding Representation.

## C   Stopwords curation

The following process manually curates the stopwords: 1. use all abstracts, do the word tokenization using NLTK, and then compute their word frequencies and rank from high to low based on the frequencies. 2. manually select stopwords from frequencies higher than 100 by checking if they are

| Approach | Examples |
|---|---|
| Baseline | **GA**: This paper presents a **machine learning** approach to the task of **word sense disambiguation** (**WSD**) based on lexical, syntactic and semantic features. Our approach relies on a rule-based classifier trained on manually annotated corpora. We show that our approach outperforms state-of-the-art **WSD** systems for English, German and Spanish. |
| | **HRA**: (Mir & Lawaye, 2020) Every language used in this world has ambiguous words. The process of analyzing the word tokens and assigning the correct meanings to the ambiguous words according to the context in which they are used is called **word sense disambiguation(WSD)**. **WSD** is a very hot research topic in Natural Language Processing. The main purpose of my research work is to tackle the **WSD** problem for Kashmiri language using Supervised **Machine Learning** Approaches |
| Frequency | **GA**:This paper presents a new approach to the automatic generation of **referring expressions** (REs) in natural language. The approach is based on the use of distributional semantics, which can be used to model the meaning of an utterance. We describe how we have constructed a data set for RE **generation** that has been manually annotated with respect to three dimensions: referential quality, fluency, and informativeness. In addition, we show how these dimensions correlate with human judgments about the quality of generated REs. |
| | **HRA**: (Marín et al., 2020) In this work we overview some of the contributions regarding the use of Fuzzy Logic in **referring expression generation**. We also discuss recent advances that can help to overcome the arguments in the literature against the use of Fuzzy Logic in Natural Language **Generation**. |
| Freq-NoLSTM | **GA**: This paper presents an overview of the current state of research in Natural Language Processing (NLP). We start with an introduction to NLP, i.e., **named entity** recognition (NER), and then go on to discuss challenges posed by NER, particularly when applied to languages other than English. Finally we conclude with some directions for future work. |
| | **HRA**: (Acharya, 2020) The paper describes how classifier model built using Conditional Random Field detects **named entities** in wet lab protocols. |
| Context | **GA**: This paper presents our work in progress for the POS tagging of Arabic documents(SBD). In this project, we propose an attentional generative model that represents text spans within chunks as Gaussian mixture functions. The main advantage is using latent **semantic information** during training and its performance improvement over other proposed methods can be demonstrated on three tasks: POS labeling with SVMs-based tagger; verb-object bracketing recognition performed by Seq2Segmenter On Evaluation Task where these techniques were combined to achieve better performances than previously published systems accuracies guarantee Accuracy FastText embedding F1 score rank s=0(grammatical error rates):.0116 vss : ·, muli =  precision nor recall pattern+1337 |
| | **HRA**: This submission is a paper that proposes an architecture for the relation extraction task which integrates **semantic information** with knowledge base modeling in a novel manner |
| Context² | **GA**: This paper presents a novel approach to the task of automatically identifying **named entities** (NEs) from text. **Named entity recognition** (NER) is one of the most important tasks in **natural language processing**. In this paper, we propose a new **neural network architecture** for NER which incorporates rich features derived from character n-grams and **word embeddings**. Our model achieves state-of-the-art results on the OntoNotes 5.0 dataset. The experimental results show that our model outperforms several baselines by a large margin. |
| | **HRA**: (Lei et al., 2020) **Named entity recognition** is a key component in various **natural language processing** systems, and **neural architectures** provide significant improvements over conventional approaches. Regardless of different **word embedding** and hidden layer structures of the networks, a conditional random field layer is commonly used for the output. This work proposes to use a neural language model as an alternative to the conditional random field layer, which is more flexible for the size of the corpus. Experimental results show that the proposed system has a significant advantage in terms of training speed, with a marginal performance degradation. |

Table 6: Example outputs for each approach. **GA**: Generated abstract from the approach. **HRA**: highest key phrases meteor matching reference abstract in the year 2020 for the generated abstract. The bold text is the matched key phrases.

just general words and do not show the key idea of the content. Finally, there are 1372 stopwords selected.

## D ADDITIONAL HUMAN EVALUATION DETAILS

We invite NLP PhD Students to do the human evaluation. They are trained using the criteria in Section 4.6. We added all of their scores up and took the average over the total full scores. The agreement score cannot be computed since it is a scoring process rather than a classification/labeling process. The average score for each criterion and total average scores for all evaluators provided enough information for model comparison.

## E SYSTEM GENERATED EXAMPLES

We give additional generated examples for each approach in Table 6.

## F FUTURE IMPROVEMENTS

The proposed task to generate future text can be improved in many aspects for future studies. First, we can apply our methods to other text generation domains such as news or tweets where content changes over time to forecast the future. Additionally, the proposed model is topic agnostic, while in the future we can improve the model to generate text about the future trends for a specific topic. The

proposed model still needs to change the model architecture and fine-tune on the temporal text data which makes it hard to apply to Large Language models (LLMs) such as GPT-4. Developing better prompting techniques with temporal information to predict future text is better for LLMs for this task. Additionally, better preprocessing of corpora for better temporal distribution can lead to better future text generation.

## G    LIMITATIONS

Our paper has several known limitations:

- The proposed task and approaches are not intended to predict totally nonexistent content in history; all of the generated content should be based on historical documents or a combination of the existing components in history.
- The content generated by this work may contain factual errors and inconsistencies, among other problems, such as citations not matching the content. The generated content will may reuse older research ideas and elements without citations, so a thorough literature review would be necessary to use this in practice.
- Our models are only designed for topic-agnostic generation, which shows general trends irrespective of the topic.
- There may be many better methods to deal with the temporal information instead of only using LSTM to model it; these methods will also be evaluated in future studies since it is out of the scope of the paper.

## H    ETHICS STATEMENT

Like any AI writing assistant, such as ChatGPT, there are dangers of applying AI generators to real-world problems without checking the output carefully. The generated content may contain old ideas that must be cited properly, and it is the responsibility of the user of any AI writing assistant to use them ethically. The ideas contained in the generated content may be useful to inspire writers, but the content generated themselves should not be directly used for real publications.