# OpenReview forum: "Future Language Modeling from Temporal Document History"
_ICLR.cc/2024/Conference — ICLR 2024 poster_

### Official Review · Reviewer_QRZD · 2023-11-03

**Soundness:** 3 good
**Presentation:** 3 good
**Contribution:** 4 excellent
**Rating:** 8
**Confidence:** 4

**Summary:**

This paper studies and introduces the task of future language modeling. This practically involves generation of text that would be written at a future time, which is not present in the pre-training of the langauge model used. This is possible through extrapolating usage from historical time intervals.

The paper introduces three methods that enhance the regular transformer-based architecture. These rely on LSTMs to model the historical frequency and predict its change with time.

A new data set and evaluation setup is proposed, where the task is to predict future abstracts from the ACL conference. Results show the proposed methods improve over all baselines.

**Strengths:**

To the best of my knowledge, the application and task are novel and interesting to study.

Three architectures are introduced, each building on each other. The effects of each component are studied and give us a sense of what is working and what is needed to improve performance.

**Weaknesses:**

More motivation and applications for this task could be provided or speculated.

The paper uses GPT-2 as the base model for all experiments. It would be better for testing robustness to potentially include a few more base models, perhaps with larger sizes or an encoder-decoder architecture for diversity.

Additional experiments and evaluation on downstream tasks can be performed using conditional generation or prompting on existing data sets with temporal dimensions e.g. the temporal NER data set (Rijhwani & Preotiuc-Pietro, ACL 2019; Luu et al NAACL 2022) or generating hashtags in future tweets using just the tweet text (similar to Preotiuc-Pietro & Cohn, EMNLP 2013). These would avoid the issues associated with evaluating generations.

Another interesting experiment to conduct would be to study prediction more into the future and quantify the expected degradation as the time window increases.

**Questions:**

Please expand on what stopword list was used, as that plays an important role in content perplexity and meteor scores.

---

> ### Author Response · Authors · 2023-11-23
> **Thank you for the detailed review and comments**
>
> Thank you for the detailed review and comments.
>
> Most of your mentioned weaknesses are what we‘d like to continue to do for future work.
>
> Answer for Questions:
>
> Since most of the stopwords lists available online are not intended for our research purposes, the stopwords are manually curated by the following process: 1. use the data in the paper, do the word tokenization using NLTK and then compute their word frequencies, and rank from high to low based on the frequencies. 2. manually select stopwords from frequencies higher than 100 by checking if they are just general words and do not show the key idea of the content. Finally, there are 1372 stopwords selected. We will add this process later in the appendix, and the stopwords list will also be released upon acceptance.

---

### Official Review · Reviewer_YEss · 2023-11-06

**Soundness:** 3 good
**Presentation:** 3 good
**Contribution:** 3 good
**Rating:** 8
**Confidence:** 3

**Summary:**

This study pioneers the task of future language modeling, aimed at generating predictive text based on historical documents. It innovates by integrating temporal biases into a pre-trained GPT-2 model, guiding the generation of future-oriented text. The models, trained on NLP paper abstracts from the ACL Anthology spanning 2009-2021, surpass traditional non-temporal models in both automated (Perplexity, METEOR) and human evaluations.

**Strengths:**

Originality:
- First to formalize future textual data prediction using temporal information.
Develops novel methods for measuring temporal dynamics in language modeling.

Quality:
- Presents a thorough structure, comparing three new models against multiple baselines.
- Demonstrates model effectiveness through careful data handling, especially distinguishing between content and non-content words during evaluation.

Clarity:
- Clearly articulates research motivations, background literature, methodology, and findings.

Significance:
- Offers significant research outcomes with implications for various applications.
- Discusses potential future applications and necessary adaptations.

**Weaknesses:**

To improve readability, you can align the organization of tables and figures more closely with their corresponding text.

**Questions:**

none

---

> ### Author Response · Authors · 2023-11-23
> **Thank you for the detailed review and comments**
>
> Thank you for the detailed review and comments.  We will fix the problem you mentioned for the presentation of the figures.

---

### Official Review · Reviewer_dCq4 · 2023-11-06

**Soundness:** 3 good
**Presentation:** 3 good
**Contribution:** 2 fair
**Rating:** 6
**Confidence:** 3

**Summary:**

The authors proposed a language model that can capture the temporal pattern of text generation. They proposed 3 model variations to attack this problem and conducted experiments over a dataset of ACL paper abstracts. Their most advanced model, the doubly contextualized model, can outperform the GPT-2 based baselines and other variations in both automatic evaluations and human evaluations.

**Strengths:**

1. Modeling of temporal patterns in LLM has not captured much attention from the community. Yet it is an important problem to look into.
2. The paper is very easy to follow. The authors did a good job of describing their ideas and approaches in simple yet accurate terms and notations.
3. The proposed models look reasonable and are proven to be effective in generating future text based on historical documents.

**Weaknesses:**

1. The proposed models are relatively simple and don't leverage the power of the most advanced LLM. Some problems the authors tried to solve, such as the gating problem in Sec 3.4 look like sth that would not be an issue to GPT-3 or other recently developed LLM as they are very effective in generating readable and coherent text. Finetuning a more powerful LLM with the latest text data seems to be a very effective way to model temporal patterns.
2. Some details and questions from the experiment were not well discussed. For example, how many raters participated in the human evaluation, what are their agreements, and how subjective are their ratings? Besides, the results in Table 4. are worth more analysis and discussion. Why do the baselines not perform well in Problem and Method? Intuitively, they should be good at generating coherent and readable content.

**Questions:**

NO

---

> ### Author Response · Authors · 2023-11-23
> **Thank you for the detailed review and comments**
>
> Thank you for the detailed review and comments. Below are our responses to weaknesses:
>
> Response to Weakness 1:
>
> The main reason we did not compare recent LLMs is potential data contamination issues.  Recent LLMs are likely to have trained on our testing dataset, and we do not know for certain if they are contaminated or not.  This make it difficult to compare recent LLMs to our approach fairly.  However our method is applicable to any LLM, and can easily be applied to recent open LLMs.
>
> The gating issue that we solve is not specific to GPT-2.  GPT-2 is already a very capable generation model, and can easily generate fluent text.  The fluency issue occurs when we apply simple bias methods (sections 3.2 and 3.3) to account for temporal information.  This causes the fluency of the model to be greatly degraded, as we show in our experiments.  The gating mechanism we introduce fixes this issue.  So we do not believe a recent LLM would have fixed this issue either, since the issue wasn’t originally present in GPT-2, but was introduce by the temporal information.  We think this is a fundamental problem of introducing temporal information to LLMs.  We will make this more clear in the text.
>
> Response to Weakness 2:
>
> We will add additional details about the number of raters and agreement for the human evaluation.
>
> The baselines do not perform well in Problem and Method because, without any constraints, they are next token prediction language models, and they hallucinate and can generate nonsense or non-relevant content (i.e. What the method talks about may far away from the problem, or an older method is used to solve the problem).  The example outputs in the appendix give a good intuition of the problems with the baseline models.  We will discuss this further in the paper.

---

### Meta-Review · Area_Chair_pfcf · 2023-12-06

**Metareview:**

This paper introduces the task of future language modeling and propose future language models. The proposed models are evaluated on the future abstract prediction task, and it outperforms the baseline non-temporal language models across all automatic evaluation metrics and human evaluation on generating content related to the future text based on temporal historical documents.

The task of future language modeling is new and interesting. The proposed language models sound reasonable and the results are promising. The paper is easy to follow.

The weaknesses of this paper lie in the evaluation part. First, only one single artificial task (i.e. future abstract prediction) is used in the experiments, which is not sufficient. Additional experiments and evaluation on real downstream tasks need to be performed. Second, only GPT-2 (which is old) is used as the base model in the experiments.  It would be better to test the robustness to include a few more base models with larger sizes.

**Justification For Why Not Higher Score:**

The evaluation is not very strong, and only one single artificial task is used for evaluation. The usability and effectiveness of the proposed model in real applications/tasks are not demonstrated.

**Justification For Why Not Lower Score:**

The problem investigated in this pape is very ininteresting and the idea is new.

---

### Decision · Program_Chairs · 2024-01-16

Accept (poster)